# Inducing Magnetic Properties with Ferrite Nanoparticles in Resins for Additive Manufacturing

**DOI:** 10.3390/ijms241411838

**Published:** 2023-07-24

**Authors:** Rocío Redón, Miriam D. Aviles-Avila, Leopoldo Ruiz-Huerta, Herlinda Montiel, Alex Elías-Zúñiga, Lucy-Caterine Daza-Gómez, Oscar Martínez-Romero

**Affiliations:** 1Instituto de Ciencias Aplicadas y Tecnología, Universidad Nacional Autónoma de México, Cd. Universitaria, Coyoacán, Mexico City 04510, Mexicoleoruiz@unam.mx (L.R.-H.); herlinda.montiel@icat.unam.mx (H.M.); cgomez.nanoscience@gmail.com (L.-C.D.-G.); 2National Laboratory for Additive and Digital Manufacturing (MADiT), Universidad Nacional Autónoma de México, Mexico City 04510, Mexico; aelias@tec.mx (A.E.-Z.); oscar.martinez@tec.mx (O.M.-R.); 3Department of Mechanical Engineering and Advanced Materials, Institute of Advanced Materials for Sustainable Manufacturing, Tecnológico de Monterrey, Av. Eugenio Garza Sada Sur 2501, Monterrey 64849, Mexico

**Keywords:** magnetic nanoparticles, cobalt ferrite, barium ferrite, additive manufacturing, VAT photopolymerization

## Abstract

Additive manufacturing and nanotechnology have been used as fundamental tools for the production of nanostructured parts with magnetic properties, expanding the range of applications in additive processes through tank photopolymerization. Magnetic cobalt ferrite (CoFe_2_O_4_) and barium ferrite (BaFe_12_O_19_) nanoparticles (NPs) with an average size distribution value (D_TEM_) of 12 ± 2.95 nm and 37 ± 12.78 nm, respectively, were generated by the hydroxide precipitation method. The dispersion of the NPs in commercial resins (Anycubic Green and IRIX White resin) was achieved through mechanochemical reactions carried out in an agate mortar for 20 min at room temperature, with limited exposure to light. The resulting product of each reaction was placed in amber vials and stored in a box to avoid light exposure. The photopolymerization process was carried out only at low concentrations (% *w*/*w* NPs/resin) since high concentrations did not result in the formation of pieces, due to the high refractive index of ferrites. The Raman spectroscopy of the final pieces showed the presence of magnetic NPs without any apparent chemical changes. The electron paramagnetic resonance (EPR) results of the pieces demonstrated that their magnetic properties were maintained and not altered during the photopolymerization. Although significant differences were observed in the dispersion process of the NPs in each piece, we determined that the photopolymerization did not affect the structure and superparamagnetic behavior of ferrite NPs during processing, successfully transferring the magnetic properties to the final 3D-printed piece.

## 1. Introduction

Ferrite nanoparticles (NPs) are magnetic materials composed of oxides that mainly consist of ferric ions. They offer several notable advantages, including suitability for higher frequencies, increased heat resistance, and improved corrosion resistance. These characteristics are crucial in electrochemical applications such as energy storage, which is closely linked to the capacitance response [1]. However, the high refractive index of ferrites (≥2.3) and their absorption in the visible and near-infrared (NIR) ranges limit their integration into optical systems. Consequently, the objective is to combine ferrites with other materials to obtain nanocomposites that mitigate these effects and maximize their potential. It has also been observed that dispersing magnetic nanoparticles in a polymeric resin prevents nanoparticle agglomeration, allowing these composite materials to be used as indicators of temperature change in magnetic resonance imaging (MRIg)-guided thermal ablations [2,3]. Nonetheless, when incorporating these ferrites into more complex systems such as composites, their magnetic properties are often lost or significantly diminished [4,5].

Materials consisting of ferrite NPs have potential use in the fabrication of scaffolds for advanced engineering of bone tissue [6], microchips [7], and force sensors [8]. In the field of biomedicine, these composites can be employed for diseases diagnosis by visualizing the spatial distribution of magnetic NPs, a technique known as magnetic particle imaging (MPI) [2]. Given the diverse applications of these compounds in various fields, it is crucial to preserve the magnetic properties of the ferrites when they are incorporated into polymeric matrices.

Another drawback arising from the use of ferrite composite-based components is the requirement for parts with defined geometries that emulate anatomical structures in certain application fields [9]. Current materials and manufacturing processes cannot achieve this, as traditional methods rely on pressure to shape specific geometries. As a result, fabricating complex structures of magnetic components, which could enhance the performance and efficiency of such devices, has been expensive or even impossible [10].

Additive manufacturing (AM), also known as three-dimensional (3D) printing, presents itself as a viable solution to this problem due its unmatched modeling capabilities and rapid prototyping. AM enables the printing of successive layers of materials based on 3D model data, allowing for the production of components with complex and intricate geometries [11]. Moreover, AM has the potential to revolutionize the design and manufacturing of power magnets. Companies such as General Electric (GE) and Siemens have already been developing AM parts using functional materials [12,13,14].

Among the various AM processes, VAT photopolymerization stands out due to its superior advantages compared to other AM methods in terms of the precision and complexity of the final parts [15]. VAT photopolymerization involves selectively curing a liquid monomer contained in a tank thorough light-initiated photopolymerization. In this process, the build platform descends one layer into the tank, and the material is hardened by light. Once the layer has hardened, the build platform descends again to continue the construction layer by layer. The energy source used for the photopolymerization can be laser, UV (UltraViolet) light (stereolithography or SLA), or direct digital light (structured light using a bitmap to project images). While this technology offers high-resolution capabilities, building materials are generally not mechanically resistant [16].

Although researchers have conducted experiments with microscale inorganic particles, the combination of AM and nanomaterials is still relatively limited in existing research. However, some studies in the field have demonstrated that the incorporation of nanostructures can have a significant impact on the final characteristics and properties of manufactured parts. By adding nanomaterials to AM print media, it becomes possible to create new composites with unique properties, thus expanding the potential application areas of AM [17,18]. Specifically, the utilization of nanofunctionalized magnetic materials opens up a greater range of possibilities and serves as a key factor in developing new applications for 3D printing, such as the production of magnetic actuators for lab-on-a-chip applications [8] or motion sensors [7].

The main objective of the current research was to produce composite materials of the nanostructure (BaFe_12_O_19_ or CoFe_2_O_4_)-resin for AM while preserving the magnetic properties of cobalt and barium ferrites in the final AM-produced pieces. These ferrites were selected as representatives of hard and soft magnets to compare their impact on the AM process and investigate the retention of magnetic properties after the AM process. This research is significant for obtaining materials and pieces that can be used in the aforementioned applications, benefiting from substantial advantages offered by AM.

## 2. Results and Discussion

### 2.1. Characterization of CoFe_2_O_4_

The XRD pattern of the CoFe_2_O_4_ obtained by the hydroxide precipitation method was analyzed to determine its crystallinity and identify different phases. Figure 1a shows the XRD pattern, which exhibits characteristic typical peaks corresponding to specific crystallographic planes, namely, (1 1 1), (2 2 0), (3 1 1), (4 0 0), (5 1 1), and (4 4 0). These peaks are indicative of the presence of the inverse spinel-type CoFe_2_O_4_ structure, in agreement with JCPDS-ICDD card 22-1086. The inverse spinel structure of CoFe_2_O_4_ has a lattice parameter a_o_ = 8.391900 Å and belongs to the space group *Fd-3m* (227), which consists of a divalent cation (Co^2+^, Fe^2+^), a trivalent cation (Fe^3+^), and a divalent anion (O^2−^). The cations A and B occupy the octahedral or tetrahedral site of the inverse spinel structure (Figure 1b). The size of nanoparticles was determined using the highest intensity signal from the diffractogram and the Scherrer equation. The analysis yielded an average diameter of 10.36 nm.

The Raman spectrum (Figure 2) shows three of the characteristic signals that are commonly reported in the literature for the inverse spinel-type cobalt ferrite (CoFe_2_O_4_) [19]. These signals are assigned for symmetric stretching at tetrahedral sites (A_1g_), symmetric bending at octahedral sites (E_g_) and antisymmetric bending at octahedral sites (T_1g_) of the crystal structure. In addition to the signals relating to the cobalt ferrite phase, there are additional signals observed around 1300 cm^−1^, which can be attributed to a contaminating phase, specifically, hematite phase (α-Fe_2_O_3_) [20].

The HR-TEM images displayed in Figure 3 reveal the presence of CoFe_2_O_4_ with a high degree of agglomeration and crystal diameters smaller than 100 nm (Figure 3a,c). A size distribution analysis of 20 NPs indicates that the majority of the particles have diameters ranging between 12 and 13 nm, with an average D_TEM_ of 12 ± 2.95 nm (Figure 3b). This size is consistent with the diameter obtained from X-ray diffraction (XRD) (10.36 nm). By performing a Fast Fourier Transform (FFT) of the selected area (Figure 3c), the interplanar distances corresponding to the crystallographic planes (2 2 2), (3 1 1), (2 2 0), and (4 0 0) of the cubic spinel CoFe_2_O_4_ crystal phase provide further evidence of the crystalline nature of the CoFe_2_O_4_ nanoparticles.

The electron paramagnetic resonance (EPR) spectrum of CoFe_2_O_4_ (Figure 4) exhibits a portion of the characteristic signal of the material, although it is not fully observable due to limitations of the equipment used. The line width (H_dc_), which represents the difference between the highest and lowest points, serves as a qualitative indicator of the dipolar magnetic interaction among the magnetic particles in the material. A wider line width indicates a higher concentration of magnetic moments and a stronger interaction among them. In the case of CoFe_2_O_4_, the observed signal is broad, suggesting a significant dipole energy between the magnetic moments of the particles and, consequently, a pronounced interaction among them.

### 2.2. Characterization of BaFe_12_O_19_ NPs

The X-ray diffraction pattern (Figure 5) shows the signals corresponding to the crystallographic planes (1 0 2), (0 0 6), (1 1 0), (0 1 7), (1 1 4), (2 0 0), (2 0 3), (2 0 5), (2 0 6), (2 0 9), (2 1 7), (2 0 11), (2 2 0), and (2 4 14) reported in ICDD-PDF N° 01-078-0133 for the crystalline phase of hexagonal barium ferrite (BaFe_12_O_19_). The hexagonal ferrite structure of BaFe_12_O_19_ has lattice parameters a = 5.829 Å, b = 5.829 Å, and c = 23.183 Å and belongs to the space group P63/mmc. The magnetic Fe^3+^ ions occupy five different crystallographic sites including octahedral sites, tetrahedral sites, and trigonal bipyramidal (TBP) sites (Figure 5b). By using the data from the most intense signal in the XRD pattern and applying the Scherrer equation, the size of the NPs was calculated, resulting in an average diameter of 49.06 nm.

The Raman spectrum (Figure 6) shows three characteristic signals reported in the literature [19] for hexagonal barium ferrite. These signals are attributed to symmetric bending in the spinel block (E_1g_) at 183 cm^−1^, bending (E_2g_) at 336 cm^−1^, and symmetric stretching at bipyramidal sites (A_1g_) of the crystal structure at 686 cm^−1^. Furthermore, a signal around 1300 cm^−1^ is also observed, which corresponds to an additional phase of hematite (α-Fe_2_O_3_) [20], similarly to the cobalt ferrite NPs mentioned earlier.

After conducting the structural characterization, microstructural features of BaFe_12_O_19_ was analyzed using TEM (Figure 7). The TEM and HR-TEM images demonstrated that the size of the NPs is smaller than 100 nm (Figure 7a,c). The particle size distribution of BaFe_12_O_19_ ranged from 32 to 42 nm, with an average D_TEM_ of 37 ± 12.78 nm (Figure 7b). This size is consistent with the size determined by X-ray diffraction, which was found to be 49.06 nm. Figure 7d corresponds to the FFT of the selected area, revealing the identification of interplanar distances in BaFe_12_O_19_. The observed interplanar spacings in these micrographs are 2.95, 2.75, and 3.85 Å, corresponding to the (1 1 0), (0 1 7), and (0 0 6) BaFe_12_O_19_ family planes, respectively. Additionally, the size distribution of the BaFe_12_O_19_ NPs was obtained from the HR-TEM images.

In the EPR spectrum for BaFe_12_O_19_ (Figure 8), a characteristic signal of the material is observed, spanning from 315.5 to 358.5 mT. The H_dc_ value of 43mT indicates the presence of magnetic dipolar interactions between the NPs in the sample. However, this magnetostatic interaction is weaker compared to the cobalt ferrite NPs previously discussed.

### 2.3. Additive Manufacturing Process

#### 2.3.1. Ferrite–IRIX White Resin

Figure 9 presents the results of additive manufactured of nanostructured samples composed of BaFe_12_O_19_ and CoFe_2_O_4_–IRIX White resin. The weight percentages (*w*%/*w*%) of the mechanochemical reactions between barium ferrite nanoparticles and White Resin, as well as those between cobalt ferrite nanoparticles and White Resin, are tabulated in Appendix A. In both cases, it is evident that the efficiency of the cylinders’ construction decreases as the concentration of magnetic NPs increases. Samples with the higher NPs concentration present a certain degree of flaking and are more prone to breakage compared to samples with a lower NPs concentration. In addition, during the initial stages of the process, we encountered some challenges which were likely attributed to the high refractive index of the ferrites (2.3 for BaFe_12_O_19_ and 2.5 for CoFe_2_O_4_) [13,14], which hindered proper UV light penetration from the equipment into the photosensitive resin, particularly at higher NP concentrations.

#### 2.3.2. Ferrite–Anycubic Green Resin

Figure 10 shows the results obtained in the AM for nanostructured samples composed of BaFe_12_O_19_ or CoFe_2_O_4_ with Anycubic Green resin. The weight percentages of the mechanochemical reactions between barium ferrite nanoparticles and Anycubic Green resin, as well as between cobalt ferrite nanoparticles and Anycubic Green resin, are provided in Appendix A. Similarly, to the IRIX resin, the efficiency of piece construction decreased as the concentration of magnetic ferrite NPs increased. When using a higher percentage of ferrite, the pieces showed a certain degree of flaking and increased flexibility compared to samples with a lower NPs percentage.

Only pieces with a concentration below 5% *w*/*w* of BaFe_12_O_19_ and below 4% *w*/*w* of CoFe_2_O_4_ could be successfully manufactured. Once again, the challenges in construction were likely attributed to the high refractive index of the ferrites (2.3 for BaFe_12_O_19_ and 2.5 for CoFe_2_O_4_) [3,21], which did not allow the UV light from the equipment to reach the photosensitive resin correctly, once concentrations above 5% were reached.

#### 2.3.3. Characterization of Composite Nanostructured Samples

The characterization of the nanostructured samples composed of ferrite-resin was carried out both before and after the AM process to determine the chemical and magnetic characteristics of the materials and determine whether magnetic properties of the NPs are preserved or altered after the process.

For the characterization of the pieces, a slice of the original piece was cut and subjected to various techniques. In the case of the Raman measurements, the laser wavelength of 532 nm, consistent with wavelength used for isolated powders of the ferrites and resins, was employed for all characterized materials.

Figure 11a shows the Raman spectra of BaFe_12_O_19_-IRIX resin nanocomposite. In addition to the signals from polymer (black spectrum in Figure 11a), the three characteristic signals reported in the literature for BaFe_12_O_19_ are observed in both spectra. In the BaFe_12_O_19_-IRIX resin spectrum, the signals are slightly shifted (barium ferrite/barium ferrite-IRIX resin: 182/185, 336/371, 686/679 cm^−1^) beyond the spectral resolution of the WITec Alpha300RA instrument (±4 cm^−1^), suggesting the existence of an interaction between the barium ferrite NPs and the resin. This interaction limits the vibrational modes in the ferrite molecules and causes their displacement towards lower energy. However, the signals for barium ferrite remain unchanged, suggesting that a change in the NPs after the AM process is not significant according to this technique.

In the case of the CoFe_2_O_4_–IRIX resin nanocomposite (Figure 11b), both the characteristic signals of the resin and those of CoFe_2_O_4_ are observed. The shifting of the ferrite signals is minimal for CoFe_2_O_4_–IRIX resin (cobalt ferrite/cobalt ferrite-IRIX resin: 314/310, 477/475, 683/679 cm^−1^). This indicates that there is not strong interaction between the ferrite and the resin once the photopolymerization process has been carried out. On the contrary, for CoFe_2_O_4_–Anycubic Green nanocomposite the displacement is significant, indicating a greater interaction between the ferrite and the resin.

By comparing the Raman spectra of the materials, it can be determined that in the case of BaFe_12_O_19_, both the nanocomposites and resin signals are present, indicating a high distribution of these nanoparticles.

Conversely, in the CoFe_2_O_4_ nanocomposites, the polymer signals are not observed in the spectra. This may indicate that CoFe_2_O_4_ has a stronger interference with the Raman laser light compared to the polymer, resulting in minimized signals from the latter in the spectrum. This observation suggests that the distribution of the CoFe_2_O_4_ NPs is predominantly on the surface of the pieces.

EPR analysis of the nanocomposites was carried out to determine the magnetic characteristics of the compounds involved in the experiment (resin, ferrites, nanostructured-composite materials) as well as the changes that the materials may have in their magnetic properties after the AM process.

Figure 12a displays the comparison of the EPR spectra of the nanostructured materials composed of BaFe_12_O_19_-IRIX White resin prior to the AM process, at various concentrations of BaFe_12_O_19_. In all the spectra containing resin, a signal located at 160 mT, assigned to the paramagnetic centers of Fe^3+^, is observed. In each graph, the characteristic signal of barium ferrite between 315.5 and 358.5 mT is observed, which suggests the presence of BaFe_12_O_19_ NPs. However, there is a variation in the linewidth of the spectra. It is known that a larger linewidth amplitude corresponds to a higher concentration of magnetic moments and a stronger interaction between them, suggesting that the particles are closer to each other. As the ferrite is “diluted” in the resin, the magnetic moments become more separated. Although there is still some interaction between them, increasing the distance between the magnetic moments leads to a narrower linewidth. The variation in the linewidths observed in the spectra (independently of the concentration of NPs) suggests that the dispersion of ferrite in the resin differs for each sample.

In Figure 12b depicts the EPR spectra of the BaFe_12_O_19_-IRIX White resin nanocomposites after photopolymerization. Similarly to samples without treatment, the signal corresponding to the magnetic dipolar interactions of BaFe_12_O_19_ is observed in all the spectra, but the linewidth varies for each graph. This variation in linewidth suggests that the dispersion of BaFe_12_O_19_ NPs is different for each sample even after the AM process.

In the case of the piece containing 0.5% *w*/*w* BaFe_12_O_19_, a signal corresponding to splitting or hyperfine interaction is observed. In other words, due to the effect of light, a rearrangement in the electronic levels occurred in the resin, enabling a direct interaction between the spin of the nucleus and the spin of unpaired electrons in the resin molecules. It is possible that the sample was exposed to light before the AM process.

By comparing the EPR results shown in Figure 12, it can be concluded that the signal corresponding to the Fe^3+^ ions and the characteristic signal of BaFe_12_O_19_ are present in all the spectra, both before and after the photopolymerization process. This indicates thar both substances—the resin and the NPs—retained their magnetic properties after the AM process.

In Figure 13a, the EPR spectra of the CoFe_2_O_4_-IRIX White resin nanocomposites before the AM process are shown. In all the spectra containing resin, the signal located at 160 mT corresponding to the paramagnetic centers of Fe^3+^ is observed. Furthermore, in each graph, the characteristic signal of cobalt ferrite is located between 100 and 460 mT, which reveals the presence of CoFe_2_O_4_ NPs in the samples. However, the linewidth corresponding to CoFe_2_O_4_ could not be obtained since the signal is incomplete due to the measurement limitations of the equipment used. Nonetheless, there is an increase in the signal intensity (in all the spectra) as the concentration of NPs increases, indicating a higher concentration of the NPs, a higher concentration of magnetic moments, and a stronger interaction between them. Essentially, the NPs are closer to each other. By “diluting” the ferrite in the resin, this leads to the separation of magnetic moments and a decrease in their interaction.

Figure 13b displays the spectra of CoFe_2_O_4_—IRIX White resin nanocomposites after photopolymerization. Once again, the signal corresponding to the Fe^3+^ paramagnetic centers of the resin is evident in all the spectra. However, the characteristic signal belonging to the NPs cannot be observed in the spectra. In the Raman spectrum for this composite nanostructured material (Figure 11), peaks associated with the ferrite NPs are observed; however, these NPs are sufficiently dispersed within the material such that the signal corresponding to the magnetic interactions between them is not visible in the spectrum.

## 3. Materials and Methods

### 3.1. Experimental Procedures

#### 3.1.1. Materials

The following compounds were obtained from a commercial supplier and used without additional purification: barium nitrate [Ba(NO_3_)_2_ (>99%)], iron nitrate nonahydrate [Fe(NO_3_)_3_)·9H_2_O (>98%)], cobalt chloride hexahydrate [CoCl_2_·6H_2_O (>98%)], iron chloride hexahydrate [FeCl_3_·6H_2_O (>97%)], anhydrous sodium hydroxide [NaOH (>97%)], and methanol [CH_3_OH (>98%)] from Sigma-Aldrich (St. Louis, MO, USA); citric acid monohydrate (C_6_H_8_O_7_·H_2_O) from Productos Químicos Monterrey (Monterrey, NL, Mexico); tetrahydrofuran [C_4_H_8_O (>99%)] and acetone (C_3_H_6_O) from Reactivos Química Meyer (Mexico City, Mexico). The products were washed using deionized triple-distilled water.

#### 3.1.2. Methods

The following methods were used in this study: The powder X-ray diffraction was run using a Siemens Equipment, D5000 Kriotalloflex Diffractometer. Raman spectra were recorded using a Raman dispersive spectrometer, specifically, the Witec model with AF confocal microscope. The laser was focused on the samples, and the scattered light was collected in a 180 °C backscattering configuration. The Raman spectra were accumulated over 25 s with a resolution of approximately 4 cm^−1^. A Nd:YVO_4_ laser operating at 532 nm (frequency doubled) was used as the excitation source. The high-resolution transmission electron microscopy (HR-TEM) analysis was conducted using a JEOL 2000F operating at an acceleration voltage of 200 kV. The electronic paramagnetic resonance (EPR) measurements were performed using JES-TE300 equipment from Jeol, Japan; the instrument operates at X band frequency, and the spectra were obtained at 300 K. The AM process using IRIX White resin was carried out on a Prefactory 3 printer from Envision TEC company. This equipment utilizes UV light with a wavelength of 385 nm. The AM process using Anycubic Green resin was performed on a Photon printer from Anycubic company. This equipment employs UV-LED light with a wavelength of 405 nm to cure the resin and fabricate the desired components.

#### 3.1.3. Synthesis of BaFe_12_O_19_ NPs

The synthesis of barium ferrite was carried out by the hydroxide precipitation method. Iron nitrate [(Fe(NO_3_)_3_·9H_2_O], barium nitrate [Ba(NO_3_)_2_], and citric acid [(C_6_H_8_O_7_)·H_2_O] were dissolved in distilled water in a molar ratio of 12:1:13. The solution was heated to 80 °C with continuous stirring for 30 min to obtain the coordination compounds (nitrate-citric acid). Subsequently, a 3 M NaOH solution was slowly added while maintaining the temperature at 80 °C with constant stirring for another 30 min to obtain the hydroxy-citrate complexes of both cations (Fe^3+^ and Ba^2+^). The resulting brown precipitate was then centrifuged and washed alternately with triple-distilled water and methanol. Afterward, it was left to dry in an atmospheric environment. Finally, the brown powders were calcinated at 750 °C for 3 h.

#### 3.1.4. Synthesis of CoFe_2_O_4_ NPs

The synthesis of cobalt ferrite was carried out by the hydroxide precipitation method. Iron chloride (FeCl_3_·6H_2_O) and cobalt chloride (CoCl_2_·6H_2_O) were dissolved in tetrahydrofuran (THF) in a molar ratio of 2:1. The solvent was partially evaporated until a green gel of ions dissolved in THF was formed. A 3 M NaOH solution was added to this gel, and the reaction mixture was heated to 80 °C with continuous stirring for two hours. This process resulted in the formation of metal hydroxides Co(OH)_2_ and Fe(OH)_3_. Once the hydroxides were synthesized, an aqueous solution of H_2_O_2_ (30%) was slowly added dropwise while stirring constantly. This step led to the formation of cobalt ferrite NPs (CoFe_2_O_4_). The resulting black precipitate was subjected to centrifugation and washed alternately with triple-distilled water and methanol. Finally, the precipitate was left to dry in an atmospheric environment.

#### 3.1.5. Obtaining Nanostructured Materials

Nanostructured materials were synthesized using mechanochemical reactions involving two photosensitive resins, namely, IRIX White (DWS Systems, Thiene, VI, Italy) and Anycubic Green (Anycubic, Shenzhen, China), along with the previously synthesized and characterized magnetic cobalt ferrite (CoFe_2_O_4_) and barium ferrite (BaFe_12_O_19_) NPs. The weight percentages (% *w*/*w*) of ferrite used in each reaction can be found in Appendix A. Each mechanochemical reaction took place in an agate mortar at room temperature, with a duration of 20 min and limited exposure to light. Subsequently, the resulting product from each reaction was transferred to amber vials and stored in a light-protected box to prevent light exposure.

#### 3.1.6. AM Process

The AM process was carried out using tank photopolymerization for both the resin and NPs—resin samples were obtained by mechanochemistry. For the nanostructured ferrite and IRIX White resin samples, cylinders shapes measuring 5 mm in diameter and 1 mm in height were manufactured using VAT photopolymerization with a UV light source emitting at a wavelength of 385 nm. Following the construction of each piece, they were washed with isopropanol and subsequently dried in a UV light chamber. For the nanostructured ferrite and Anycubic Green resin samples, the manufacturing process involved obtaining complex pieces, specifically, the base of a chess tower. This was achieved through VAT photopolymerization using a UV light source with a wavelength of 405 nm. After the completion of the AM, the pieces were washed with isopropanol and left to dry in an atmospheric environment.

## 4. Conclusions

The synthesis of nanometric-sized barium and cobalt ferrites with superparamagnetic behavior was achieved through the precipitation method using hydroxides. The magnetic behavior of the samples was confirmed by electron paramagnetic resonance analysis, which exhibited characteristic signals (315–358 mT for BaFe_12_O_19_ and 100–450 mT for CoFe_2_O_4_) associated with dipolar interactions between magnetic NPs. It was observed that magnetic interaction between CoFe_2_O_4_ NPs was stronger compared to that between BaFe_12_O_19_ NPs.

The photopolymerization process was successfully carried out at low concentrations of NPs (up to 4.0% *w*/*w* for CoFe_2_O_4_ and 5.0% *w*/*w* for BaFe_12_O_19_). However, at higher concentrations, the refractive index of the ferrites hindered the formation of solid pieces as the UV light was unable to effectively penetrate the photomonomer.

The Raman spectroscopy confirmed the presence of crystalline phases of both ferrites in the manufactured pieces, indicating that no apparent chemical changes occurred during the AM process. Furthermore, a stronger interaction between BaFe_12_O_19_ and the IRIX White resin was observed, suggesting a closer association between the ferrite NPs and the resin matrix. Finally, based on the EPR results, the manufactured pieces by the AM process retained the magnetic properties of the NPs used, indicating that the magnetic properties of the NPs were preserved during the photopolymerization process. However, the variation in line width in the EPR spectra indicated that the dispersion of NPs within each piece was different, suggesting differences in their spatial distribution within the matrix.

## Figures and Tables

**Figure 1 ijms-24-11838-f001:**
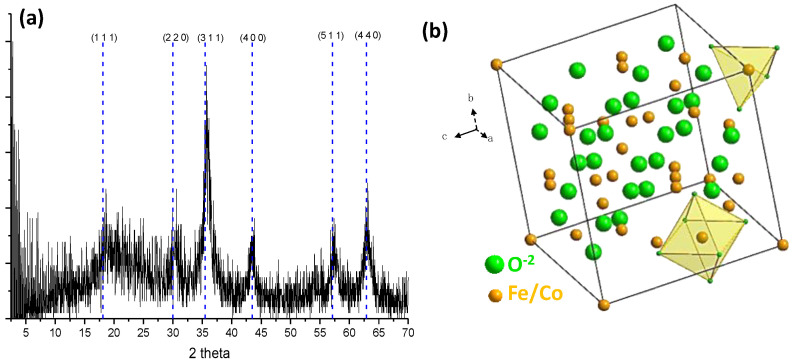
(**a**) The XRD pattern and (**b**) the representation of inverse spinel-type CoFe_2_O_4_ unit cell obtained with the hydroxide precipitation method.

**Figure 2 ijms-24-11838-f002:**
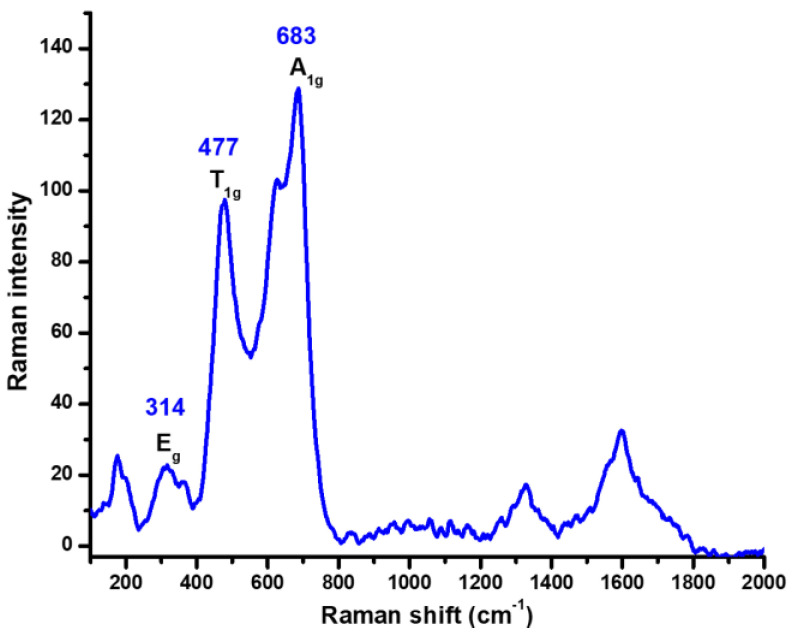
Raman spectra of CoFe_2_O_4_ NPs obtained with the hydroxide precipitation method.

**Figure 3 ijms-24-11838-f003:**
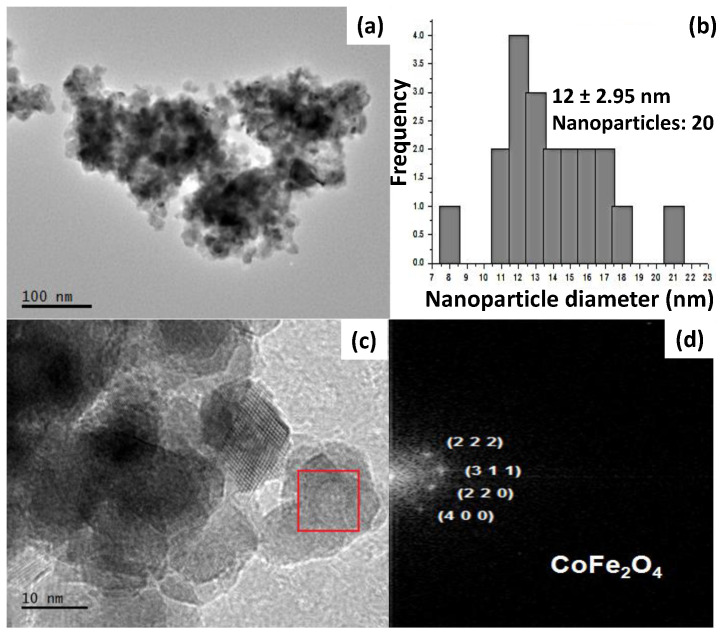
(**a**) HR-TEM image of CoFe_2_O_4_ obtained with the hydroxide precipitation method and (**b**) the corresponding histogram, representing a narrow size distribution of the NPs. (**c**) An enlarged HR-TEM image with red box depicting the area used for (**d**) FFT.

**Figure 4 ijms-24-11838-f004:**
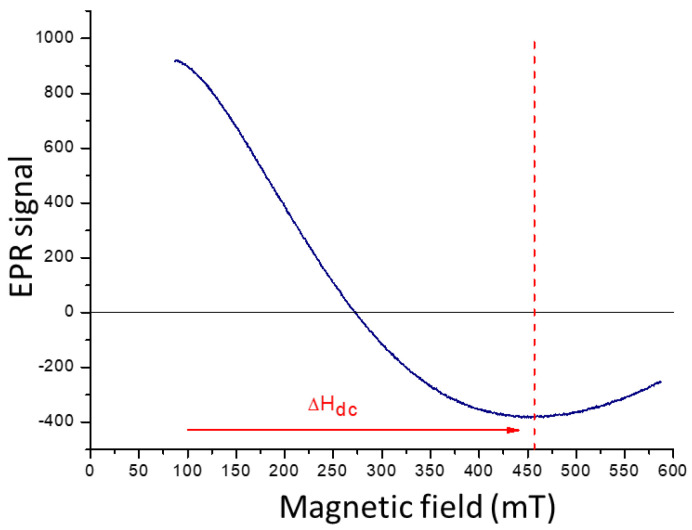
EPR spectra of CoFe_2_O_4_ obtained with the hydroxide precipitation method.

**Figure 5 ijms-24-11838-f005:**
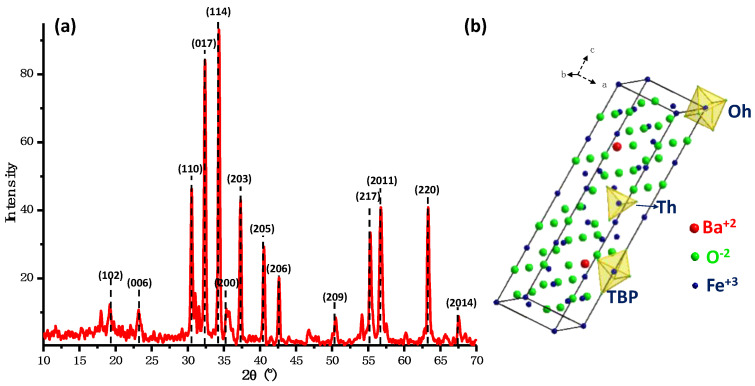
(**a**) The XRD pattern and (**b**) the representation of hexagonal BaFe_12_O_19_ unit cell obtained with the hydroxide precipitation method.

**Figure 6 ijms-24-11838-f006:**
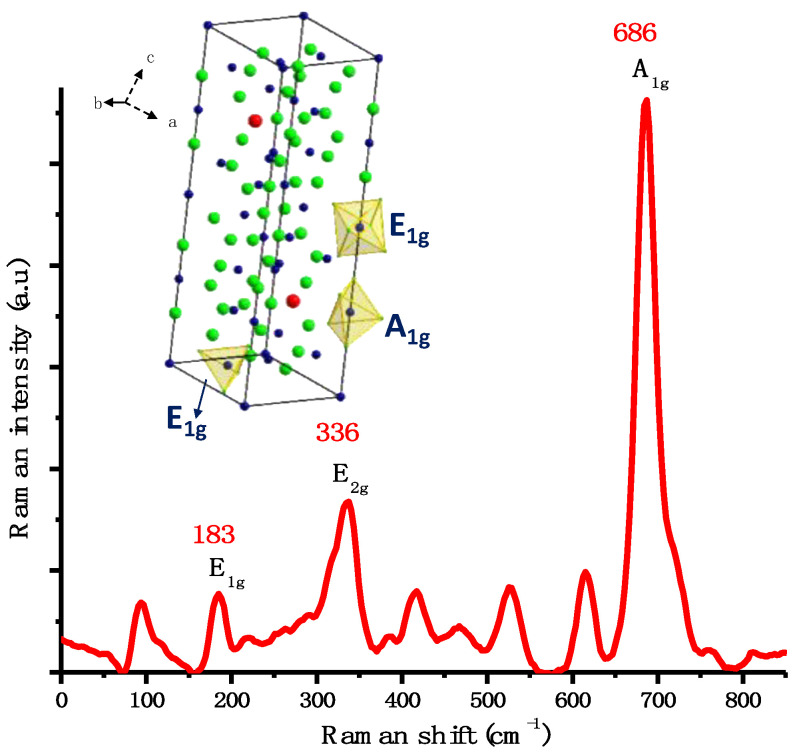
Raman spectra of BaFe_12_O_19_ NPs obtained with the hydroxide precipitation method and the representation of hexagonal BaFe_12_O_19_ unit cell with the corresponding vibrational modes.

**Figure 7 ijms-24-11838-f007:**
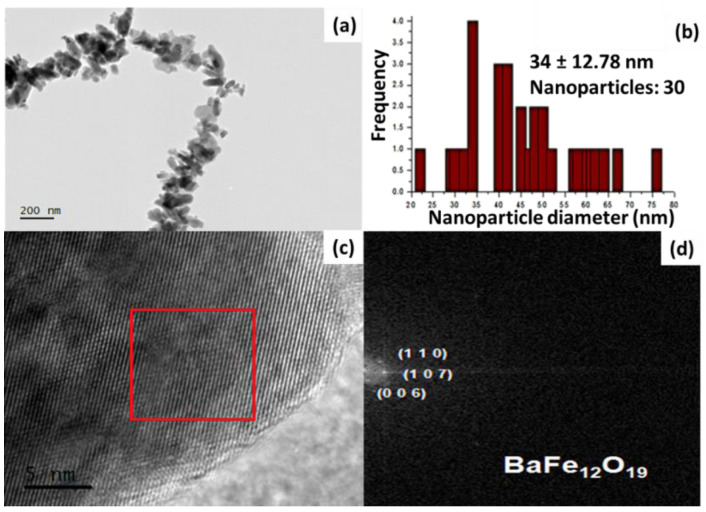
(**a**) HR-TEM image of BaFe_12_O_19_ obtained with the hydroxide precipitation method and (**b**) the corresponding histogram, representing a narrow size distribution of the NPs. (**c**) An enlarged HR-TEM image with red box depicting the area used for (**d**) FFT.

**Figure 8 ijms-24-11838-f008:**
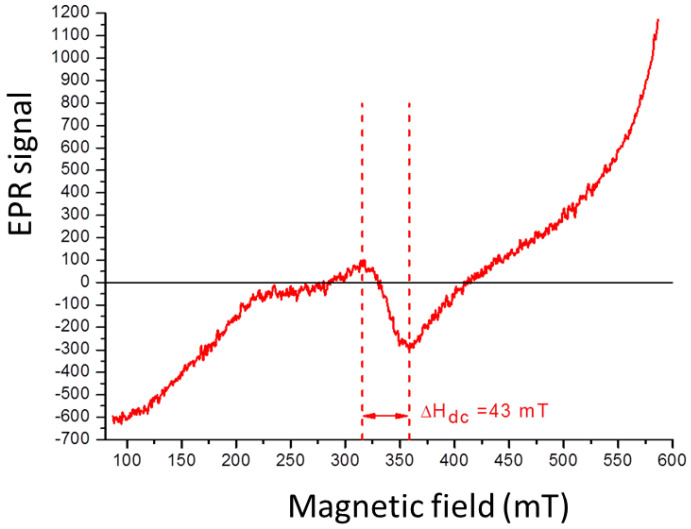
EPR spectra of BaFe_12_O_19_ obtained with the hydroxide precipitation method.

**Figure 9 ijms-24-11838-f009:**
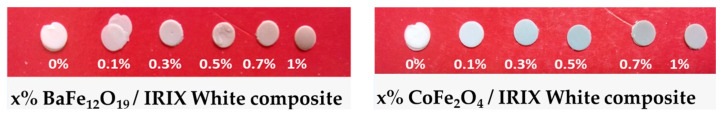
Cylinders built by AM by photopolymerization from two different nanostructured ferrites composed of IRIX White resin with different percentages of ferrite.

**Figure 10 ijms-24-11838-f010:**
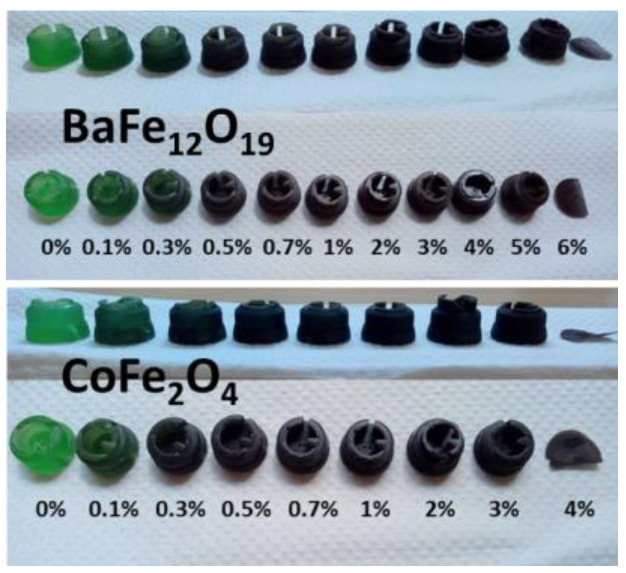
Pieces built by AM by photopolymerization from two different nanostructured ferrites composed of Anycubic Green resin with varying percentages of ferrite (0–6% for BaFe_12_O_19_ (**top**) and 0–4% for CoFe_2_O_4_ (**bottom**)).

**Figure 11 ijms-24-11838-f011:**
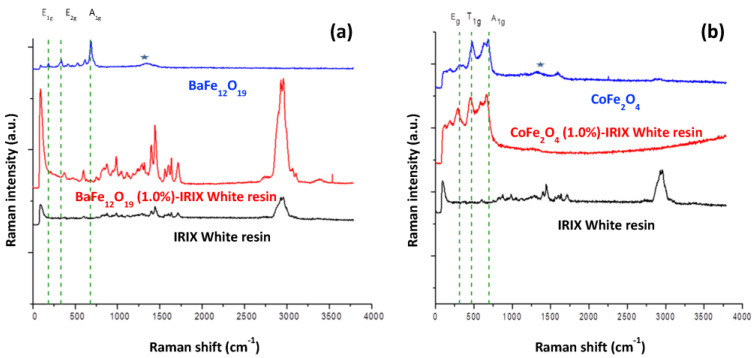
Comparison of Raman spectra of (**a**) BaFe_12_O_19_ and (**b**) CoFe_2_O_4_. Raman spectrum of barium or cobalt ferrite only (blue spectrum), nanostructured material composed of barium or cobalt ferrite-IRIX White resin (red spectrum) at 1% *w*/*w*, and IRIX White resin only (black spectrum). ^★^ a-Fe_2_O_3_ hematite contamination.

**Figure 12 ijms-24-11838-f012:**
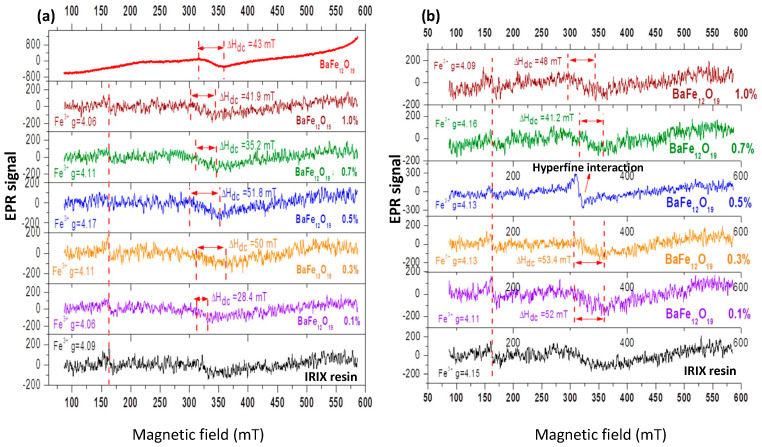
Electron paramagnetic resonance spectra taken at 300 K of BaFe_12_O_19_–IRIX White resin samples (**a**) before and (**b**) after the AM process.

**Figure 13 ijms-24-11838-f013:**
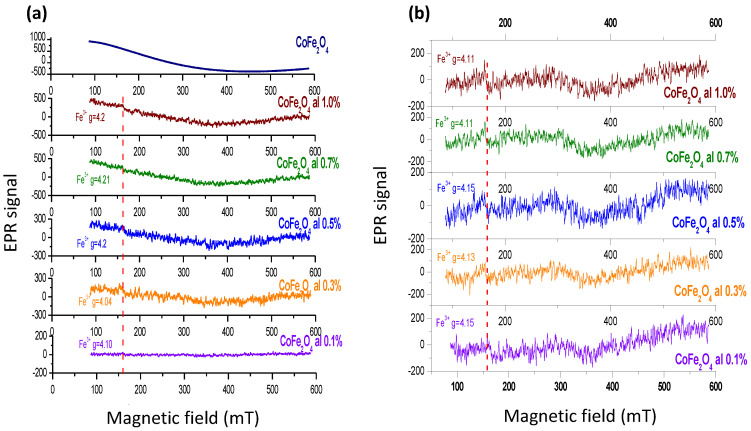
Electron paramagnetic resonance spectra taken at 300 K of CoFe_2_O_4_-IRIX White resin samples (**a**) before and (**b**) after the AM process.

## Data Availability

Not applicable.

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
