# Peer review of "Inducing Magnetic Properties with Ferrite Nanoparticles in Resins for Additive Manufacturing"

_ijms, 2023, doi:10.3390/ijms241411838_

Round 1
Reviewer 1 Report
Review Report:
In their manuscript Redón et al. demonstrated the synthesis and characterization of materials with magnetic iron nanoparticles (NPs) for additive manufacturing (AM). The two NPs used for the present study are CoFe2O4 and BaFe12O19 with average sizes 12 and 37 nm, respectively. The dispersed NPs on two resins were made to undergo photopolymerization at lower NP concentrations. Through the implementation of Raman and EPR spectroscopy they showed that there is essentially no change both in chemical and magnetic properties during the photopolymerization process.
The manuscript is well-written, thorough, and easy to follow. However, there are a few minor modifications required before final publication. I have used the following abbreviations, P-page number and L-line number to point out my comments.
Specific Comments:
1. P2-L65: It is not needed to capitalize the additive manufacturing.
2. P2-L96: Punctuation issue at the beginning of this sentence.
3. P3-L99: I could not understand what does “Provided a reference” stand for.
4. P4-L108: Is it possible to estimate the percentage of the contaminating phase.
5. P5-L159: The average size of the NP is not consistent with the figure.
6. I was wondering at what temperature the EPR data were taken.
7. P8-L221-227: The entire paragraph needs to be modified as it is not consistent with the figure (Figure 11).
In addition, the line colors are not properly defined in the figure caption of Figure 11.
8. Both Figure 12 and Figure13 are hard to follow. I encourage the authors to provide a more resolved EPR spectra.

Author Response
Reviewer #3 report: In their manuscript Redón et al. demonstrated the synthesis and characterization of materials with magnetic iron nanoparticles (NPs) for additive manufacturing (AM). The two NPs used for the present study are CoFe2O4 and BaFe12O19 with average sizes 12 and 37 nm, respectively. The dispersed NPs on two resins were made to undergo photopolymerization at lower NP concentrations. Through the implementation of Raman and EPR spectroscopy they showed that there is essentially no change both in chemical and magnetic properties during the photopolymerization process. The manuscript is well-written, thorough, and easy to follow. However, there are a few minor modifications required before final publication. I have used the following abbreviations, P-page number and L-line number to point out my comments. Specific Comments:
- P2-L65: It is not needed to capitalize the additive manufacturing.
Thank you for this observation, but as far as we know, in the available literature we haven´t found the development of nanocomposites based on polymeric resins with incorporated ferrites thus, this we would like to emphasize this work as a first approach to the development of parts obtained with additive manufacturing.
- P2-L96: Punctuation issue at the beginning of this sentence.
Thank you, we have extensively checked the grammar of the article, the manuscript is now edited accordingly.
- P3-L99: I could not understand what does “Provided a reference” stand for.
We are so sorry for this mistake; it was a note to the authors to include a related reference.
- P4-L108: Is it possible to estimate the percentage of the contaminating phase.
As we did not know that the contaminating will appear, we did not include any standard to evaluate the concentration of it, nevertheless, by taking only the data of the signals in the spectra, the contamination is around 10%, although, the amount of ferrite used in the composite is 1%, thus the contaminating phase is imperceptible.
- P5-L159: The average size of the NP is not consistent with the figure.
I am sorry, but in the text is written that “most of the NPs between 12 and 13 nm in diameter with an average DTEM of 12 ± 2.95 nm (Figure 3b). This size coincides with that obtained by X-ray diffraction (XRD) (10.36 nm).” The histogram presents these values and the micrography depicted in Figure 3c, shows some particles around these values, the size bar is of 10 nm. It might be possible that you did not see the number in the size bar?
- I was wondering at what temperature the EPR data were taken.
Thank you, it was taken at room temperature, we already include the information in the Figure caption.
- P8-L221-227: The entire paragraph needs to be modified as it is not consistent with the figure (Figure 11). In addition, the line colors are not properly defined in the figure caption of Figure 11.
Thank you we have extensively checked the grammar of the article with a specialized paper editor company, the manuscript is now edited accordingly.
- Both Figure 12 and Figure13 are hard to follow. I encourage the authors to provide a more resolved EPR spectrum.
We apologize for this inconvenient, we already improve the quality of the Figures.
Thank you for your extensive and careful review of the paper, we hope now it is in better shape to be publish.

Reviewer 2 Report
In the manuscript, the authors have explored the possibility of including magnetic nanoparticles in resins for additive manufacturing processes, to obtain magnetic samples with arbitrary shapes. In this aim, two kinds of magnetic ferrite nanoparticles have been mixed with two commercial resins at several different concentrations and tested. The samples have been characterized by X-Ray diffractometry, HR transmission electron microscope, Raman Spectroscopy, and EPR electron paramagnetic resonance.
The EPR showed that pieces manufactured by the additive manufacturing process have magnetic properties and these magnetic properties apparently do not change during the photopolymerization process.
The investigation process is interesting and timely. The work approach and methods used are valid.
The obtained results are interesting but their presentation has to be improved. The main points to be addressed are the following:
Firstly, the relevance of producing magnetic pieces by additive manufacturing should be better stated and examples of applications have to be given.
On page 4, line 125 (pag.9, line 283): please, specify what limitations of the equipment do not allow the full characterization of the sample.
On page 5, Fig. 6; in the sketch, the labels are not congruent with the text.
On page 4, Fig. 3, and page 6, Fig. 7: in the caption, refer the shown figures to a,b,c,d index.
On page 7. line 191 … : The commercial name of the resin is “Anycubic”: Use this name instead of “ANY cubic” all along the text.
On page 8, Fig.11: specify the content of (a) and (b) referring to these indices. The label “iris” have to be corrected to “irix”. The label “Ag1” should be correct to “A1g”. Two “*” are added to the plots : what is their meaning? Check the text of the caption and specify the line color for each spectrum.
On page 9, Fig.12 a signal located at 160 mT occurs in the bare resin spectra. This signal is “assigned to the paramagnetic centers of Fe3+”, as stated on page 8, lines 245-246. Fe3+ is present also in the bare resin? Please comment on this aspect..
Correct labels in the figure, where “resina” is reported instead of “resin”
In Fig.11, the g-factor value is about 4. Typically, the value assigned to Fe3+ is of order of 2 in Barium ferrite (see for instance Y. Chen et al, J. Appl. Phys. 99, 08M904 ). Please, comment on this point.
On page 10, line 312: correct “lase” to “laser”.
Ref. 17; add the page number.
For these reasons I recommend the publication of the manuscript after a major revision of the text.
Author Response
Reviewer #2 report: In the manuscript, the authors have explored the possibility of including magnetic nanoparticles in resins for additive manufacturing processes, to obtain magnetic samples with arbitrary shapes. In this aim, two kinds of magnetic ferrite nanoparticles have been mixed with two commercial resins at several different concentrations and tested. The samples have been characterized by X-Ray diffractometry, HR transmission electron microscope, Raman Spectroscopy, and EPR electron paramagnetic resonance.
The EPR showed that pieces manufactured by the additive manufacturing process have magnetic properties and these magnetic properties apparently do not change during the photopolymerization process.
The investigation process is interesting and timely. The work approach and methods used are valid.
The obtained results are interesting but their presentation has to be improved. The main points to be addressed are the following:
Firstly, the relevance of producing magnetic pieces by additive manufacturing should be better stated and examples of applications have to be given.
Thank you for your suggestion, we add more examples of AM application in the introduction.
On page 4, line 125 (pag.9, line 283): please, specify what limitations of the equipment do not allow the full characterization of the sample.
We appreciate the comment, we have extended the explanation about the limitations coming from the high refractive indices of the ferrites in the construction of the pieces (page 10)
On page 5, Fig. 6; in the sketch, the labels are not congruent with the text.
We apologize for this error and thank you for the observation, we have corrected both the image and the description in the text.
On page 4, Fig. 3, and page 6, Fig. 7: in the caption, refer the shown figures to a,b,c,d index.
Thank you for the annotation, we modified the legend of the figures 3 and 7, including the indices (a), (b), (c) and (d).
On page 7. line 191 … : The commercial name of the resin is “Anycubic”: Use this name instead of “ANY cubic” all along the text.
Thank you, we have standardized the name of the resin throughout the text.
On page 8, Fig.11: specify the content of (a) and (b) referring to these indices. The label “iris” have to be corrected to “irix”. The label “Ag1” should be correct to “A1g”. Two “*” are added to the plots : what is their meaning? Check the text of the caption and specify the line color for each spectrum.
We appreciate your observation. We modified the legend of the Figures 11, including the indices (a) and (b). Besides, we have changed label “iris” to “IRIX” and label “Ag1”.
The “*” corresponds to the hematite phase (α-Fe2O3), this is mentioned in the text of Figure 6, and we also included the description on the Figure 6 caption.
The text of the title and the specification of the color of the lines for each spectrum were corrected.
On page 9, Fig.12 a signal located at 160 mT occurs in the bare resin spectra. This signal is “assigned to the paramagnetic centers of Fe3+”, as stated on page 8, lines 245-246. Fe3+ is present also in the bare resin? Please comment on this aspect..
Correct labels in the figure, where “resina” is reported instead of “resin”
Thank you for the observation, we have modified Figure 12, and translating the words from Spanish to English. The signal at 160 mT is assigned to Fe3+, and appears in spectrum of the bare resin due to small contamination of the resins but, it got an increment on the composite sample, thus we conclude that the observed effect is due to the ferrite nanoparticles.
In Fig.11, the g-factor value is about 4. Typically, the value assigned to Fe3+ is of order of 2 in Barium ferrite (see for instance Y. Chen et al, J. Appl. Phys. 99, 08M904). Please, comment on this point.
Thank you, depending on the nature of the material the g-factor may vary, for example in this reference V N Vasyukov 2011 J. Phys.: Conf. Ser. 324 012024, DOI: 10.1088/1742-6596/324/1/012024, denote two possible lines in the EPR spectra at ~2.0 and 4.0 (J. Wakabayashi; Paramagnetic Resonance Spectrum of Fe3+ in Calcite. J. Chem. Phys. 15 April 1963; 38 (8): 1910–1912. https://doi.org/10.1063/1.1733895). Other references also report the g factor around 4.0o (H. Hollis Wickman, Melvin P. Klein, D. A. Shirley; Paramagnetic Resonance of Fe3+ in Polycrystalline Ferrichrome A. J. Chem. Phys. 15 March 1965; 42 (6): 2113–2117. https://doi.org/10.1063/1.1696253)
On page 10, line 312: correct “lase” to “laser”.
We appreciate the comment, we made the corresponding correction.
Ref. 17; add the page number.
We sorry for this mistake, we add the page and number to reference 17 (now ref 18)
For these reasons I recommend the publication of the manuscript after a major revision of the text.
Thank you for your extensive and careful review of the paper, we hope now it is in better shape to be publish.

Reviewer 3 Report
In the present paper, the authors report the procedures for manufacturing composite materials using two kinds of ferrite nanoparticles and two photosensitive resins. The prepared samples were characterized by using different experimental techniques. The authors evidenced that the ferrite nanoparticles embedded in resins maintain the magnetic properties and they are not seemingly modified during the photopolymerization.
The topic addressed by the authors is interesting given the relevance of the use of composite materials with magnetic properties in additive manufacturing, but relevant efforts should be made for improving the manuscript.
The title should be more incisive, the term “material” is too generic. The 36-42 lines in the Introduction are not very clear. What is the exact meaning of the sentence “Among its greatest advantages are applicability to higher frequencies, higher heat resistance and higher corrosion resistance”? The list of the proposed biomedical applications is not very pertaining. The following lines 44-52 present some sentences that are a kind of repetition.
Many Figure captions are not correct or unclear (see Figures 1,3,5,9,10,11). Figure S1 is missing.
As far as concerns Raman spectroscopy measurements, the spectral resolution of the used equipment should be given in order to make a significant comparison among the positions of the peaks in different spectra, as done in Figure 11 and related discussion. Please notice that the use of decimal digits in wavenumber peak has no meaning from a physics point of view if the experimental set-up does not have a suitable spectral resolution.
In the description of experimental instruments on lines 311-318 please give more technical details about them and their producers. Some sentences are not clear.
It will be beneficial for increasing the relevance of the prepared composite materials if the authors evidence the advantages of their materials in comparison with other ferrite nanoparticle composite materials. Please report some actual examples of the use of the described materials in the biomedical field or in other applicative frameworks.
Please notice that a large part of the Supplementary Materials cited at the end of the manuscript is not available.
In my opinion, the paper cannot be accepted for publication in its present form and major revisions are required.
A careful revision of the English language is strictly required. The manuscript should be also amended for various misprints.
Author Response
Reviewer #1 report: In the present paper, the authors report the procedures for manufacturing composite materials using two kinds of ferrite nanoparticles and two photosensitive resins. The prepared samples were characterized by using different experimental techniques. The authors evidenced that the ferrite nanoparticles embedded in resins maintain the magnetic properties and they are not seemingly modified during the photopolymerization.
The topic addressed by the authors is interesting given the relevance of the use of composite materials with magnetic properties in additive manufacturing, but relevant efforts should be made for improving the manuscript.
Thank you for your valuable comment. We take into consideration your comments to improve the manuscript.
Mayor corrections were underlined in yellow in the manuscript, minor corrections were only changed.
The title should be more incisive, the term “material” is too generic. The 36-42 lines in the Introduction are not very clear. What is the exact meaning of the sentence “Among its greatest advantages are applicability to higher frequencies, higher heat resistance and higher corrosion resistance”? The list of the proposed biomedical applications is not very pertaining. The following lines 44-52 present some sentences that are a kind of repetition.
Many Figure captions are not correct or unclear (see Figures 1,3,5,9,10,11). Figure S1 is missing.
Thank you for the observations, we have extensively checked the grammar of the article with a specialized paper editor company, thus the manuscript is now edited accordingly, and the figure captions had been rewritten. And the title is now different according to the reviewer's suggestion.
We would like to apologize for the error, the S1 is not included, since the S1 figure is already included in the Figure 11 a and b as the black spectra, thus we decided not to include the figure separately, but we forgot to eliminate the note on the manuscript.
As far as concerns Raman spectroscopy measurements, the spectral resolution of the used equipment should be given in order to make a significant comparison among the positions of the peaks in different spectra, as done in Figure 11 and related discussion. Please notice that the use of decimal digits in wavenumber peak has no meaning from a physics point of view if the experimental set-up does not have a suitable spectral resolution.
Thank you for the observation, you are right, and we apologize for this absence of information, we include now the spectral resolution (the resolution of the Almega used in the present research is of 4cm-1), thus the manuscript is now corrected accordingly.
In the description of experimental instruments on lines 311-318 please give more technical details about them and their producers. Some sentences are not clear.
It will be beneficial for increasing the relevance of the prepared composite materials if the authors evidence the advantages of their materials in comparison with other ferrite nanoparticle composite materials. Please report some actual examples of the use of the described materials in the biomedical field or in other applicative frameworks.
We include now some extra examples, and we have explained better the experimental instruments.
Please notice that a large part of the Supplementary Materials cited at the end of the manuscript is not available.
Thank you for the observation. We apologize for this mistake; the supplementary material is now included.
In my opinion, the paper cannot be accepted for publication in its present form and major revisions are required.
Thank you for your extensive review of the paper, we hope now it is in better shape to be publish.
Comments on the Quality of English Language
A careful revision of the English language is strictly required. The manuscript should be also amended for various misprints.
Thank you, we have extensively checked the grammar of the article with a specialized paper editor company, the manuscript is now edited accordingly.

Round 2
Reviewer 2 Report
The Authors suitably modify the manuscript taking into account almost all comments, and overcoming the main criticisms.
However, some corrections have been missed in the revision process. In Fig. 6 the modes are marked by “Eg1” instead of “E1g”. In the caption of Fig. 7, refer to the plots with a,b,c, d index, as done in the case of Fig. 3.
Thus, I recommend the publication of the manuscript after fixing these points.
Author Response
Reviewer #2: The Authors suitably modify the manuscript taking into account almost all comments and overcoming the main criticisms.
However, some corrections have been missed in the revision process. In Fig. 6 the modes are marked by “Eg1” instead of “E1g”. In the caption of Fig. 7, refer to the plots with a,b,c, d index, as done in the case of Fig. 3.
Thank you very much for the observation, we are so sorry for this oversight, thus we have changed the image of Figure 6 with the correct information, and the title of Figure 7 has been changed.
Thus, I recommend the publication of the manuscript after fixing these points.
Thank you again for your profound review of the paper, we hope that now it is in good shape to be published.

Reviewer 3 Report
In my opinion, the authors tried to improve their manuscript, but the present version is still unsatisfactory.
As far as concerns the title, “Inducing magnetic properties with ferrite nanoparticles in resins for additive manufacturing" could be a better choice.
The 36-64 lines of the introduction are still confusing. In my previous report, I asked about the meaning of the sentence “Among its greatest advantages are applicability to higher frequencies, higher heat resistance and higher corrosion resistance”. This sentence is still present, but the authors didn’t give any explanation. The authors should try present in an orderly manner the applications of materials composed of ferrite NPs in the different fields. In the present version of the introduction, the authors present applications in the different biomedical and industrial framework without clearly explaining the different aims.
The final sentence of the Introduction “…. this ferrites were chosen as an example of hard and soft magnets to compare their effect on the AM process as well as to study the prevalence of the magnetic properties after the AM process” is not very clear and doesn’t give an indication about the relevance and novelty of the reported results.
As far as the technical details about the Raman spectrometer, no changes are present in the paper. In addition, if the authors know that the spectral resolution of their apparatus is 4 cm-1, why do they indicate peak positions in figures 2 and 6 using decimal digits (314.2, 477.2 cm-1)?
Please notice that the Figure 6 caption is not complete.
Please take into account that the current name is “Raman spectroscopy” and not “Raman shift spectroscopy” (see the Abstract and the Conclusions).
Please notice that now there is Table S1 available on the website, but it is not mentioned in the manuscript.
The English language still requires a careful revision
Author Response
Reviewer #3: In my opinion, the authors tried to improve their manuscript, but the present version is still unsatisfactory.
As far as concerns the title, “Inducing magnetic properties with ferrite nanoparticles in resins for additive manufacturing" could be a better choice.
Thank you for the suggestion, we take it and change the title to the proposed one.
The 36-64 lines of the introduction are still confusing. In my previous report, I asked about the meaning of the sentence “Among its greatest advantages are applicability to higher frequencies, higher heat resistance and higher corrosion resistance”. This sentence is still present, but the authors didn’t give any explanation.
Thanks again for the observation. We modified the introduction. Regarding the sentence “Among its greatest advantages are applicability to higher frequencies, higher heat resistance and higher corrosion resistance”, these characteristics are very important in electrochemical applications such as energy storage, since, for example, the working frequency is related to the capacitance response, a very important property in energy storage. The high resistance to corrosion is also a very important factor, since it guarantees a greater utility of materials based on ferrites.
The authors should try present in an orderly manner the applications of materials composed of ferrite NPs in the different fields. In the present version of the introduction, the authors present applications in the different biomedical and industrial framework without clearly explaining the different aims.
The final sentence of the Introduction “…. these ferrites were chosen as an example of hard and soft magnets to compare their effect on the AM process as well as to study the prevalence of the magnetic properties after the AM process” is not very clear and doesn’t give an indication about the relevance and novelty of the reported results.
We have reorganized the introduction, and we have incorporated sentences where the different objectives are explained, including the final sentence.
As far as the technical details about the Raman spectrometer, no changes are presented in the paper. In addition, if the authors know that the spectral resolution of their apparatus is 4 cm-1, why do they indicate peak positions in figures 2 and 6 using decimal digits (314.2, 477.2 cm-1)?
We appreciate the comment, the technical details of the Raman equipment are updated and presented in the Materials and Methods section (2.1.2. Methods). Also, you are right and we forgot to change the Raman values that now they are fitted in the text and in the corresponding Figures.
Please notice that the Figure 6 caption is not complete.
Thank you for the observation and we sorry for the omission, the Figure 6 caption is now complete.
Please take into account that the current name is “Raman spectroscopy” and not “Raman shift spectroscopy” (see the Abstract and the Conclusions).
Thank you for the precision, we changed The lines where Raman shift spectroscopy appeared.
Please notice that now there is Table S1 available on the website, but it is not mentioned in the manuscript.
Comments on the Quality of English Language
The English language still requires a careful revision
Thank you for the observation, we have extensively re-checked the grammar of the article with a specialized paper editor company, the manuscript is now edited accordingly.
Thank you again for your extensive and profound review of the paper, we hope that now it is in good shape to be published.

Round 3
Reviewer 3 Report
In my opinion, the paper can now be accepted for publication.
A few misprints are still present but they can be eliminated during the proof editing stage.